# Specific Cellular and Humoral Immune Responses to the Neoantigen RBD of SARS-CoV-2 in Patients with Primary and Secondary Immunodeficiency and Healthy Donors

**DOI:** 10.3390/biomedicines11041042

**Published:** 2023-03-28

**Authors:** Kauzar Mohamed Mohamed, Kissy Guevara-Hoyer, Carlos Jiménez García, Laura García Bravo, Adolfo Jiménez-Huete, Antonia Rodríguez de la Peña, Beatriz Mediero Valeros, Cristina Cañizares Velázquez, Esther Culebras López, Noemí Cabello, Vicente Estrada, Ángel L. Corbí, Miguel Fernández-Arquero, Alberto Ocaña, Alberto Delgado-Iribarren, Mercedes Martínez-Novillo, Estefanía Bolaños, Eduardo Anguita, Ascensión Peña, Celina Benavente, Javier David Benítez Fuentes, Pedro Pérez Segura, Silvia Sánchez-Ramón

**Affiliations:** 1Department of Immunology, Laboratory Medicine Institute (IML) and Fundación para la Investigación Biomédica del Hospital Clínico San Carlos (IdISSC), Hospital Clínico San Carlos, Calle Profesor Martín Lagos SN, 28040 Madrid, Spain; kauki96@gmail.com (K.M.M.);; 2Department of Immunology, Ophthalmology and ENT, School of Medicine, Complutense University, 28040 Madrid, Spain; 3Department of Neurology, Hospital Ruber Internacional, 28034 Madrid, Spain; 4Department of Microbiology, IML and IdISSC, Hospital Clínico San Carlos, 28040 Madrid, Spain; 5Unit of Infectious Diseases, Department of Internal Medicine, Hospital Clínico San Carlos, Calle Profesor Martín Lagos SN, 28040 Madrid, Spain; 6Centro de Investigaciones Biológicas (CSIC), C./Ramiro de Maeztu, 9, 28040 Madrid, Spain; 7Clinical Analysis Department, Laboratory Medicine Institute (IML) and Fundación para la Investigación Biomédica del Hospital Clínico San Carlos (IdISSC), Hospital Clínico San Carlos, Calle Profesor Martín Lagos SN, 28040 Madrid, Spain; 8Department of Hematology, Hospital Clínico San Carlos, IML, IdISSC, Calle Profesor Martín Lagos SN, 28040 Madrid, Spain; 9Department of Medical Oncology, Hospital Clínico San Carlos, Calle Profesor Martín Lagos SN, 28040 Madrid, Spain; 10Department of Clinical Immunology, Hospital Universitario Clínico San Carlos and IdISSC, Calle Profesor Martín Lagos SN, 28040 Madrid, Spain

**Keywords:** primary immunodeficiencies, secondary immunodeficiencies, COVID-19, SARS-CoV-2 cellular response, SARS-CoV-2 humoral response, CVID, antibody deficiency disorders

## Abstract

Patients with antibody deficiency disorders, such as primary immunodeficiency (PID) or secondary immunodeficiency (SID) to B-cell lymphoproliferative disorder (B-CLPD), are two groups vulnerable to developing the severe or chronic form of coronavirus disease caused by SARS-CoV-2 (COVID-19). The data on adaptive immune responses against SARS-CoV-2 are well described in healthy donors, but still limited in patients with antibody deficiency of a different cause. Herein, we analyzed spike-specific IFN-γ and anti-spike IgG antibody responses at 3 to 6 months after exposure to SARS-CoV-2 derived from vaccination and/or infection in two cohorts of immunodeficient patients (PID vs. SID) compared to healthy controls (HCs). Pre-vaccine anti-SARS-CoV-2 cellular responses before vaccine administration were measured in 10 PID patients. Baseline cellular responses were detectable in 4 out of 10 PID patients who had COVID-19 prior to vaccination, perceiving an increase in cellular responses after two-dose vaccination (*p* < 0.001). Adequate specific cellular responses were observed in 18 out of 20 (90%) PID patients, in 14 out of 20 (70%) SID patients and in 74 out of 81 (96%) HCs after vaccination (and natural infection in some cases). Specific IFN-γ response was significantly higher in HC with respect to PID (1908.5 mUI/mL vs. 1694.1 mUI/mL; *p* = 0.005). Whereas all SID and HC patients mounted a specific humoral immune response, only 80% of PID patients showed positive anti-SARS-CoV-2 IgG. The titer of anti-SARS-CoV-2 IgG was significantly lower in SID compared with HC patients (*p* = 0.040), without significant differences between PID and HC patients (*p* = 0.123) and between PID and SID patients (*p* =0.683). High proportions of PID and SID patients showed adequate specific cellular responses to receptor binding domain (RBD) neoantigen, with a divergence between the two arms of the adaptive immune response in PID and SID patients. We also focused on the correlation of protection of positive SARS-CoV-2 cellular response to omicron exposure: 27 out of 81 (33.3%) HCs referred COVID-19 detected by PCR or antigen test, 24 with a mild course, 1 with moderate symptoms and the remaining 2 with bilateral pneumonia that were treated in an outpatient basis. Our results might support the relevance of these immunological studies to determine the correlation of protection with severe disease and for deciding the need for additional boosters on a personalized basis. Follow-up studies are required to evaluate the duration and variability in the immune response to COVID-19 vaccination or infection.

## 1. Introduction

The coronavirus disease 2019 (COVID-19) pandemic has posed a threat to public health, especially in specific groups at risk for severe COVID-19 (obesity, diabetes, asthma or chronic lung disease and sickle cell disease, including immunocompromised patients). Clinical presentations of COVID-19 vary quite widely in immunodeficient patients, ranging from asymptomatic to severe acute respiratory syndrome (SARS) [1,2]. The most clinically prevalent primary immunodeficiency (PID) is common variable immunodeficiency (CVID), a predominantly antibody deficiency estimated in 1 of 25,000 to 50,000 individuals [3]. According to the European Society for Immunodeficiencies (ESIDs) criteria, CVID is characterized by a marked reduction in immunoglobulin IgG and IgA with or without reduced IgM levels, as well as reduced frequencies of switched memory B cells and/or diminished vaccine antibody responses [4,5].

On the other hand, patients with hematological cancer have been defined as the most vulnerable group for severe COVID-19-related morbidity and mortality [6]. In this secondary immunodeficiency (SID) of predominantly antibody defect, both the underlying disease and the immunosuppressive treatment contribute to the immunodeficiency. Disease-related factors that contribute to immunodeficiency include B-cell dysfunction leading to hypogammaglobulinemia, advanced age and comorbidities, which are known factors for severe COVID-19 [7,8].

Failure to produce specific antibody responses to pathogens in PID and SID has raised concerns about the risk for severe or prolonged infection with SARS-CoV-2 and the potential benefits of immunization against SARS-CoV-2 in terms of protective specific T-cell responses in these patient populations [9,10]. Interferon (IFN)-γ is a key player in driving antiviral cellular immunity through the activation of, for instance, macrophages and triggering specific cytotoxic immunity [11]. In vitro production of IFN-γ can be used as a reliable read-out of specific memory T cells in circulation [12]. 

As with many other infections, both natural immunization through infection and vaccination can develop long-term immunity against SARS-CoV-2 [13,14]. With the ongoing rollout of COVID-19 vaccinations, B- and T-cell responses to SARS-CoV-2 in healthy convalescent or vaccinated donors are well described to date, whereas fewer data exist in PID and SID patients [15,16,17,18,19]. Reliable diagnostic assays are critical for evaluating the relation of specific cellular and humoral responses to SARS-CoV-2 vaccines in immunodeficient patients and for establishing their correlates of protection [20,21].

The purpose of this study is to compare cellular and humoral responses in PID patients and in SID patients to hematological cancer with a group of healthy controls, and to better define the overall efficacy of host immune responses after new exposure to SARS-CoV-2 infection.

## 2. Material and Methods

### 2.1. Study Design

The total study population was composed of 121 subjects: 20 PID patients (aged 17 to 74 years, 14/20 females), 20 SID patients with to hematological cancer (aged 49 to 85 years, 15/20 females) and 81 healthy controls (aged 23 to 83 years, 60/81 females). Samples were collected before the administration of the first vaccine dose in 10 of the PID patients. Patients and HCs were vaccinated at weeks 0 and 4. Additionally, for all patients, samples were collected between 3 and 6 months after the second vaccine dose or after recovering from COVID-19 as part of the follow-up of patients in the Clinical Immunology Unit. Patients on immunoglobulin replacement therapy (IgRT) were vaccinated between IgRTs. The study was conducted according to the guidelines of the Declaration of Helsinki, and approved by the Institutional Review Board. The study was approved by the Ethics Committee of the Hospital Clínico San Carlos (20/243-E_BS). Written informed consent for clinical data and blood sample collection was waived given the emergency of the current pandemic. All participating donors were included in this report, and no subject was excluded.

### 2.2. Evaluation of SARS-CoV-2 Cellular Response

T-cell response to SARS-CoV-2 was measured using Quan-T-Cell SARS-CoV-2 kit (Euroimmun, Lübeck, Germany) within 16 h of blood withdrawal and was analyzed on a Triturus analyzer (Grifols S.A., Barcelona, Spain). Human lithium-heparin plasma, obtained after stimulation using the SARS-CoV-2 IGRA stimulation tube set, was diluted 1:5 in the sample buffer. Afterward, 100 μL of each calibrator, control and diluted sample was added to high-binding 96-well ELISA plates pre-coated with monoclonal anti-IFN-γ antibodies. After 2 h of incubation at room temperature (RT), plates were washed 5 times with 350 μL of wash buffer. Subsequently, 100 μL of biotin-labeled anti-interferon-gamma antibody was added into each of the microplate wells and incubated for 30 min at RT. After following washes as described above, 100 μL of peroxidase-labeled streptavidin was added and incubated for 30 min at RT. After five additional washes with wash buffer, 100 μL of 3,3′,5,5′-tetramethylbenzidine/peroxide (TMB/H_2_O_2_) was added to each well, incubated for 20 min, and the absorbance was read at 450 nm 30 min after the stop solution was added (sulfuric acid). The interpretation of SARS-CoV-2 IFN-γ antibody testing was as follows: <100 mUI/mL = negative, ≥100 to <200 = borderline, ≥200 = positive. 

### 2.3. Evaluation of SARS-CoV-2 Humoral Response

Humoral immune response was assessed by a highly sensitive and specific chemiluminescent microparticle immunoassay (SARS-CoV-2 IgG II Quant assay on an ARCHITECT analyzer; Abbott Laboratories, Chicago, IL, USA), according to the manufacturer’s instructions. The assay detects IgG antibodies against the S1 subunit receptor binding protein (RBD) of the SARS-CoV-2 spike protein. A value ≥50 arbitrary units per milliliter (AU/mL) was considered evidence of vaccination response.

### 2.4. Statistical Analysis

Microsoft Excel (v.14.1.0), GraphPad Prism software (version 8.1.0) and R software (version 4.0.4) were used for descriptive and statistical data analysis. Categorical variables were compared using Fisher’s exact test or chi-squared test, as appropriate. The difference in IFN-γ levels between clinical groups (PID, SID and HC) was analyzed with both bivariable (Kruskall–Wallis and Mann–Whitney U) and multivariable (logistic regression) tests. The dependent variables in logistic regression were coded as values greater than or less than the median of IFN-γ levels. The characteristics of the variables did not allow us to use linear regression. Values were expressed as mean ± standard deviation (SD) or median (Q1–Q3), and *p*-values of less than 0.05 were considered significant. 

## 3. Results

### 3.1. Epidemiological and Immunological Characteristics of the Study Population

To understand immune responses to SARS-CoV-2 in patients with predominantly antibody deficiencies, 20 PID patients who fulfilled CVID diagnosis according to ESID criteria, 20 patients with SID to B-cell lymphoproliferative disorder (B-CLPD) and 81 healthy controls (HCs) were consecutively assessed. The study population characteristics are shown in Table 1 and Appendix A. All PID patients had suffered from recurrent respiratory tract infections and had decreased IgG, IgA and/or IgM levels before IgRT. There were three CVID patients with high or normal serum IgM, which were suspected of hyper-IgM but did not show IgM genetic variants. A total of 6 out of 20 PID patients (30%) had lymphoproliferative manifestations: 1 with a current clinical course of large granular lymphocytic (LGL) leukemia and 5 cases (25%) had clinical signs of autoimmune disease. Five PID patients (25%) had lymphocytopenia, one of which after rituximab treatment years after CVID diagnosis. All but 1 patient received IgRT with an improvement in infections, 6 patients (30%) were treated with subcutaneous (SC) IgRT and 13 patients (65%) were treated with intravenous (IV) IgRT. 

Among patients diagnosed with SID to B-CLPD, the most frequent cancer was non-Hodgkin’s lymphoma (n = 12, 60%), chronic lymphocytic leukemia (n = 5, 25%), monoclonal gammopathy of undetermined significance (n = 2, 10%) and multiple myeloma (n = 1, 5%). A total of 15 out of 20 patients (75%) with SID were on IgRT (all IV).

### 3.2. SARS-CoV-2 History

CVID patients were vaccinated with the first and second vaccine dose from April to June, 2021, depending on the vaccine brand received. According to the Spanish SARS-CoV-2 vaccine schedule, 17 patients (85%) received two doses of the mRNA-1273 vaccine (Moderna), 2 patients (10%) received the mRNA vaccine BNT162b2 (Pfizer/Biontech) and 1 patient (5%) received the adenovirus-vectored vaccine ChAdOx1 nCoV-19 (AstraZeneca). Five CVID patients (25%) had already presented with COVID-19, with asymptomatic (n = 2), mild (n = 2) and severe (n = 1) clinical courses (Figure 1A).

Among patients with SID, 11 cases (55%) received the mRNA-1273 vaccine (Moderna), and the remaining 9 patients (45%) received the mRNA vaccine BNT162b2 (Pfizer/Biontech). They received the two doses from January to June, 2021. A total of 3 out of the 20 SID patients presented with COVID-19, with mild (n = 2) and severe (n = 1) clinical courses (Figure 1B).

With respect to the population of 81 HCs, 70 controls (86.4%) were vaccinated with the first and second doses from January to June, 2021, according to the Spanish SARS-CoV-2 vaccine calendar. A total of 51 HCs (72.9%) received the mRNA vaccine BNT162b2 (Pfizer/Biontech), 12 (17.1%) received the mRNA-1273 vaccine, 5 (7.1%) received the adenovirus-vectored vaccine ChAdOx1 nCoV-19 (AstraZeneca) and 2 cases (2.9%) received a single dose of the adenovirus-vectored vaccine Ad.26.COV2.S (Janssen). The remaining 11 HCs had not received any vaccine at the time of the study, 8 out of 11 experienced COVID-19 and 3 were neither vaccinated nor with known COVID-19 to determine the specificity of the test. Globally, 31 HCs (39.2%) had presented with COVID-19, with asymptomatic (n = 3), mild (n = 23) and severe (n = 5) clinical courses (Figure 1C).

We did not sequence the specific variants in our patients. However, the Microbiology Dept. of our hospital performed sequencing of variants in random COVID-19 patients to ascertain the predominant SARS-CoV-2 variant at every wave. At the time of our study, the omicron variant was the predominant variant.

### 3.3. SARS-CoV-2 T-Cell Responses in PID and SID Patients

Pre-vaccine-specific SARS-CoV-2 cellular responses were evaluated in 10 PID patients with median (IQR) IFN-γ levels of 80.5 (0–270) mUI/mL, which is significantly lower with respect to post-vaccination levels (*p* < 0.001) (Figure 2B). A total of 4 out of these 10 PID patients (40%) who had COVID-19 prior to vaccination (1 patient with bilateral pneumonia and the rest with mild symptoms) showed higher specific anti-SARS-CoV-2 IFN-γ production than those without previous disease, as expected, 270 (68.7–1375) mUI/mL vs. 1.1 (0–17) mUI/mL. One additional PID patient without known disease, whose parents in close contact had had COVID-19, showed baseline indeterminate results, suggesting asymptomatic COVID-19.

Post-vaccine results showed a positive cellular response in 18 out of 20 (90%) PID patients with median (IQR) IFN-γ levels of 1694.1 (651.5–1856.5) mUI/mL and in 74 out of 81 (96%) HCs with median (IQR) IFN-γ levels of 1908.5 (1149.5–2001.0) mUI/mL, with significant differences (*p* = 0.005) (Figure 2B). Specific anti-SARS-CoV-2 IFN-γ responses in the remaining two PID patients were indeterminate, with both having lymphocytopenia. One PID patient showed panhypogammaglobulinemia and slight T CD8^+^ lymphocytopenia (195 cell/uL) (see Appendix A). PID11 displayed a low increase in IFN-γ production (from 0 to 99 mUI/mL) post-vaccine, whereas the other PID patient developed T CD8^+^ lymphocytopenia (196 cell/uL) after rituximab treatment.

With respect to the 20 SID patients, specific cellular responses showed median (IQR) IFN-γ levels of 1877.9 (167–1937) mUI/mL without differences with HC (*p* = 0.215). Additionally, no differences were observed between PID and SID groups (*p* = 0.371). Positive T-cell responses were observed in 14 out of 20 (70%) SID patients. Three SID patients displayed borderline specific anti-SARS-CoV-2 IFN-γ responses, with no response in the remaining three patients (see Appendix A). 

The difference in IFN-γ levels between clinical groups (PID, SID and HC) remained statistically significant after adjusting for age, sex, time since last exposure to SARS-CoV-2 antigens (vaccination or natural infection) and past history of natural SARS-CoV-2 infection with logistic regression (LR test, *p* = 0.016). An inspection of the model showed that this difference depended on the difference between the PID and HC groups (Wald test, *p* = 0.013), while it was not significant for the contrast between the SID and HC groups (Wald test, *p* = 0.760). 

### 3.4. SARS-CoV-2-Specific T-Cell Responses in Healthy Donors

As mentioned previously, 74 out of 81 (96%) HCs had positive cellular responses. Among the seven patients who had not achieved positive IFN-γ production, three were unvaccinated and without known COVID-19 to ensure the specificity of the study. The remaining four presented borderline cellular responses without a medical history suggestive of immunodeficiency. HCs were divided into three subgroups: SARS-CoV-2 vaccination (HC-vaccine); naturally immunized by infection (HC-nat-immun); or both (HC-hybrid). Regarding the cellular immune responses, 39 out of 42 (93%) HC-vaccine patients, all 8 HC-nat-immun (100%) patients and 27 out of 28 (96.4%) HC-hybrid patients showed positive IFN-γ levels. The median (IQR) IFN-γ levels are as follows: 1863 (1064–1995) mUI/mL for HC-vaccine, 1601.1 (945–1.944) mUI/mL for HC-nat-immun and 1964.5 (1879–2059) mUI/mL for HC-hybrid (Figure 3). Specific anti-SARS-CoV-2 IFN-γ levels were higher in the hybrid HC subgroup without significant differences with the other subgroups, but a trend was observed (*p =* 0.097). A greater variability in specific anti-SARS-CoV-2 cellular responses was observed in the HC-vaccine subgroup, despite the narrow time frame (3–6 months) with respect to HC-nat-immun (3–18 months). Unfortunately, only three controls had a response to natural infection and were unvaccinated, showing high responses up to 18 months, but the sample size does not permit any statistical analysis. 

### 3.5. SARS-CoV-2 Antibody Responses

SARS-CoV-2 humoral testing showed adequate antibody responses in 80% of PID patients with median (IQR) anti-spike IgG levels of 2015 (51–5611) UA/mL. PID6 was the only patient who failed to produce positive adaptive immune responses. As mentioned previously, she had B (31 cells/µL) and CD8^+^ T lymphocytopenia. All SID patients tested for antibodies (17/20, 85%) developed positive humoral responses with a median (IQR) of 465.7 (238–1984). All HCs tested for humoral responses (14/81, 17%) presented adequate humoral responses with a median (IQR) of 4392 (2528–13,384) UA/mL, which is significantly higher than SID patients (*p* = 0.040), and without significant differences between PID and HC patients (*p* = 0.123) and PID and SID patients (*p* = 0.683) (Figure 4). Interestingly, no correlation was observed between specific cellular responses and specific humoral responses to SARS-CoV-2. 

### 3.6. SARS-CoV-2 Infection Follow-Up and Correlate for Protection

Our study population was followed up after the omicron wave in Spain to analyze whether high cellular IFN-γ titers correlated with protection against subsequent exposure to SARS-CoV-2 infection. A total of 3 out of 20 PID patients (PID1, PID7 with previous COVID-19 plus two doses of vaccine and PID11 with indeterminate IFN-γ titers after two doses of vaccine) had positive PCR for SARS-CoV-2 during the sixth wave of the coronavirus in Spain, all of them related to mild symptoms. Among SID patients, only one patient (SID20) experienced asymptomatic COVID-19 after a routine RT-PCR testing positive. We further tested the correlate for protection of positive SARS-CoV-2 cellular response to omicron exposure: 27 out of 81 (33.3%) HCs referred COVID-19 detected by PCR or antigen test, 24 with mild symptoms, 1 with a moderate course and the remaining 2 with bilateral pneumonia that were treated in an outpatient basis. The two HC subjects with bilateral pneumonia (HC39 and HC51) (11.1%) showed high IFN-γ titers of 2083 mUI/mL and 1958 mUI/mL, respectively. None of them required hospitalization and are currently without sequels of the disease. From the IGRA study to date, 17 subjects presented with mild COVID-19. Intriguingly, all controls (n = 3), SID patients (n = 6) and PID patients (n = 8) with COVID-19 had presented low or negative IGRA values and had received one (n = 4) or two boosters (n = 6).

## 4. Discussion

In the present study, we assessed the immunogenicity of COVID-19 infection and/or vaccination in the setting of PID and SID and compared it to a large healthy control group. We show a detectable specific SARS-CoV-2 cellular response in 90% of PID patients, 70% of SID patients and 96% of HCs. Specific humoral responses were detected in all SID and HC patients, whereas only 80% of PID patients displayed positive RBD-specific antibodies, much greater than expected in primary antibody production deficiencies. This latter result is in line with other cohort reports showing that the majority of PID patients are able to respond to SARS-CoV-2 vaccines or infection [18,19,22]. However, other authors have reported low humoral responses in PID patients after a SARS-CoV-2 vaccine [23]. Our findings in SID patients are in agreement with a study by Mairhofer et al. [24]. The authors found detectable antibody levels in 57.8% of patients, while we found detectable levels in 100% of patients in our series, which might be due to the fact that their patients were under cancer treatment, whereas our patients were not on active anticancer treatments. The differences in responses between PID and SID patients may be due to deeper cellular immunodeficiency in SID patients. The titers of specific antibody responses were low in SID patients for the same reason, despite the fact that patients were previously immunocompetent and may show secondary responses to coronavirus.

This study has several limitations that should be considered regarding its relatively small sample size for the immunodeficiency groups and the intrinsic heterogeneity of the SID group, although all were classified as B-CLPD. Another potential limitation was the lack of corresponding data on humoral immunity in all patients due to availability issues to better clarify the correlation with cellular responses. We did not dilute the supernatants with IFN-γ concentrations above 2000 mIU/mL. Additionally, the IGRA test does not differentiate between CD4- and CD8-SARS-CoV-2-specific T cells. Further studies are needed to understand the degree of each cell subset participation in the response after infection or vaccination. Moreover, there is sample-to-sample variation in the absolute lymphocyte counts when fixing blood samples in the presence of S peptides. However, we were looking for easy, specific and reliable tests to study cellular immune response to SARS-CoV-2 and the correlate of protection, which is a major goal in health care right now. Therefore, we present this research as a pilot exploratory study.

On the other hand, our study has several strengths. Even though our PID patients had failed to mount a humoral response to polysaccharide or tetanus vaccines, they mostly presented an adequate humoral response to RNA-based SARS-CoV-2 vaccines and infection. From the PID diagnosis viewpoint, RBD ARN immunization does not seem to reliably discriminate between patients and healthy controls as an evaluation test. Furthermore, the cellular response observed in PID patients with antibody deficiency was reassuring and supports vaccination in this population. Similarly, the IFN-γ T-cell responses of patients with PID to the influenza vaccine have been described before [25,26]. In those PID patients whose T-cell response was measured before vaccination, the vaccine enhanced cellular responses induced by natural infection, which suggests a prominent boosting effect of vaccination in all convalescent PID patients [27]. Despite the high percentage of PID responders, IFN-γ levels were significantly lower than HC. We noticed that individuals who had borderline or negative cellular responses had T lymphocytopenia, which could probably explain, in part, their in vitro low IFN-γ production [28]. 

In the case of HC, the hybrid subgroup presented the highest specific anti-SARS-CoV-2 IFN-γ levels, which might indicate that the cellular response induced by natural infection was significantly enhanced by subsequent vaccination [29,30]. All naturally immunized HCs disclosed positive and long-lasting cellular responses (up to 18 months now), highlighting the relevance of monitoring anti-SARS-CoV-2 IFN-γ production to help immunization decisions. According to Krüttgen et al., we observed interindividual variability in cellular responses among the vaccination HC subgroup: we can distinguish high responders and low responders to the different types of SARS-CoV-2 vaccines [31]. Low responders might need additional booster doses, as established for other vaccines [32]. Parallel to the differences observed in cellular responses, we also observed high heterogeneity in specific antibody levels. According to Pilz et al., more than a third of our study population has been infected by SARS-CoV-2, most with mild symptoms [33], while 11.1% presented with bilateral pneumonia despite high IFN-γ levels that may probably account for virus escape to the omicron variant. In the follow-up study after IGRA analysis to date, an additional 14% of subjects presented with mild symptomatic COVID-19 despite most of them having received booster doses. All of these patients and controls that presented with symptomatic COVID-19 were those with low or negative IGRA values, probably reflecting that high IGRA responses may be mostly associated with asymptomatic virus exposure. Our results warrant further studies in a larger series of patients to evaluate whether there exists a correlation between cellular and humoral responses, and the best correlate of protection.

In summary, a two-dose SARS-CoV-2 vaccine appears to be beneficial in primary and secondary immunodeficient patients. Our observations underline the need for additional boosters in low responders to SARS-CoV-2 vaccines to achieve a better immune response against the virus. Our data from the vaccinated group with adequate cellular responses show a very good profile of protection against the omicron variant. Immunization after a natural infection seems to elicit positive and long-lasting adaptive immunity responses, without severe infection after re-exposure. Further analysis is ongoing to assess, in the near future, the duration of these immune responses to SARS-CoV-2, the effects of the third dose in immunodeficient patients and to determine the best correlate for protection after infection and after vaccination in our study populations. 

## Figures and Tables

**Figure 1 biomedicines-11-01042-f001:**
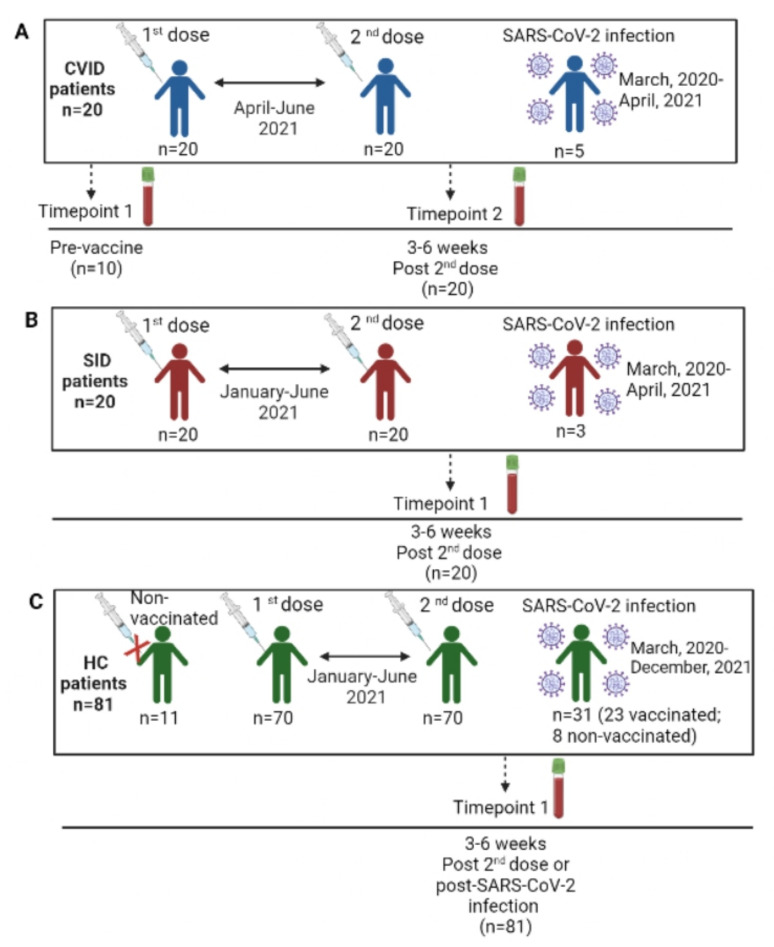
Schedule of vaccination and SARS-CoV-2 infection of our study population. (**A**) CVID patients. (**B**) SID patients. (**C**) HC patients.

**Figure 2 biomedicines-11-01042-f002:**
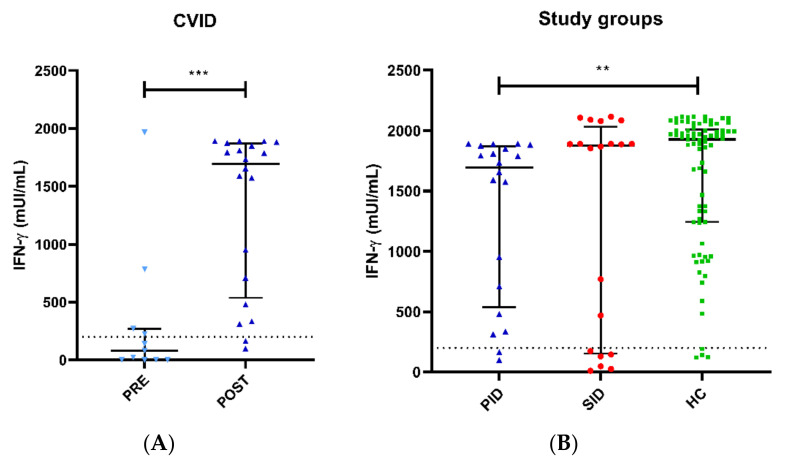
Specific SARS-CoV-2 cellular responses measured by IGRA. (**A**) Pre-vaccine and post-vaccine anti-SARS-CoV-2 IFN-γ levels in PID patients, with titers significantly lower before the vaccine administration with respect to SARS-CoV-2 post-vaccination levels (*p* < 0.001). (**B**) Patients with PID are shown in blue and SID in red compared to immunocompetent controls in green. PID patients exhibited significantly lower IFN-γ anti-SARS-CoV-2 titers than HC (*p* = 0.005). ** *p* < 0.01, *** *p* < 0.001.

**Figure 3 biomedicines-11-01042-f003:**
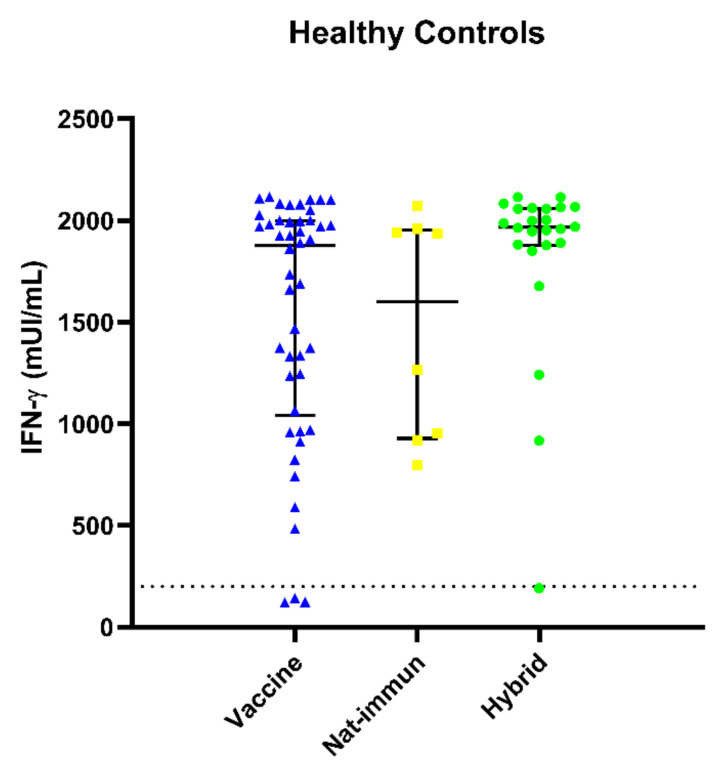
Specific anti-SARS-CoV-2 IFN-γ levels in healthy control (HC) subgroups (vaccine showed in blue, natural immune response in yellow and hybrid in green) assessed by IGRA. There was great variability in the HC-vaccine subgroup in comparison with the other subgroups, and the highest titers were observed in the HC-hybrid subgroup.

**Figure 4 biomedicines-11-01042-f004:**
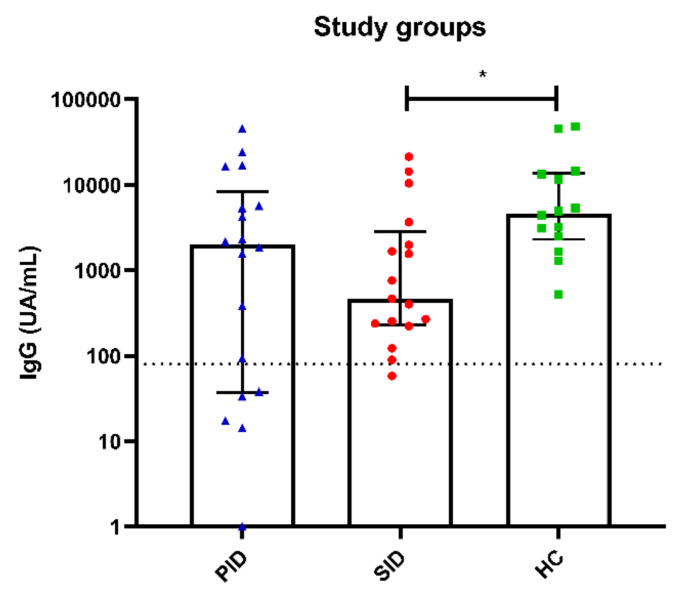
Antibody anti-SARS-CoV-2 responses measured by chemiluminescent microparticle immunoassay in the three study groups, PID represented with blue triangles, SID with red circles and HC with green squares. SID to B-CLPD mounted significantly lower IgG titers than HC. * Significant differences (*p* < 0.05).

**Table 1 biomedicines-11-01042-t001:** Epidemiological features of the three study groups.

	CVID No. = 20	SID No. = 20	Healthy Controls No. = 81
M/F	6/14 (30%/70%)	5/15 (25%/75%)	21/60 (26%/74%)
Age (years)	49.8 ± 16.4	67.9 ± 8.8	48.1 ± 15.6
IgG at diagnosis (mg/dL)	464 ± 276 499 (269–595)	720 ± 1106 369 (162–816)	NA
IgA at diagnosis (mg/dL)	41 ± 44 40 (0–51)	56 ± 64 25 (12–104)	NA
IgM at diagnosis (mg/dL)	64 ± 112 17 (5–52)	32 ± 45 11 (8–43)	NA
CD4^+^ T-lymphocytes (/uL)	659 ± 288 598 (431–870)	900 ± 650 725 (436–1222)	NA
CD8^+^ T-lymphocytes (/uL)	574 ± 414 530 (257–682)	751 ± 370 709 (440–1047)	NA
IgRT	SCIG 6/20 (30%) IVIG 13/20 (65%)	IVIG 15/20 (75%)	NA
COVID-19 prior to vaccine	5/20 (25%)	3/20 (15%)	31/81 (38%)

M: male; F: female. Results are expressed as no. (%) or mean + ESM, median (Q1–Q3).

## Data Availability

The datasets generated and analyzed during the current study are available from the corresponding author on reasonable request.

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
