# Peer review of "Specific Cellular and Humoral Immune Responses to the Neoantigen RBD of SARS-CoV-2 in Patients with Primary and Secondary Immunodeficiency and Healthy Donors"

_biomedicines, 2023, doi:10.3390/biomedicines11041042_

Round 1

Reviewer 1 Report

This paper mainly discusses the cellular and humoral immunity against SARS-CoV-2 in patients with immunodeficiency. As most studies investigate SARS-CoV-2 immunity in healthy donors, this study would be of interest.  Contrary to my expectation, according to the conclusion of this paper, patients with immunodeficiency do not seem to have impaired immunity compared to healthy donors after vaccination. 

Merits: The introduction is well-written with a clear logic. The discussion part covers the major conclusions of this paper and the data are well-presented.

Drawbacks: 

1. Please provide a data plot (with individual values of scattered plot and mean/std) for Table.1 on the antibody levels, CD4 and CD8 T cells (can be in supplementary information). The data in this table looks questionable, especially the IgM level, where average (64) exceeds maximum (52)

2. Use a high-resolution (better .svg file, or directly exported as 600dpi jpeg/png format) image for all your figures. The y-axis of your figure should be IFN-γ not γIFN

3. The IFN-γ level of your samples look questionable. In this paper, 10.1016/j.jcv.2022.105098   they used the same kit and the result looks more plausible (max. 10000 or more). The maximum value of your IFN-γ looks similar, with a plateau of ~2000, which might be due to inappropriate dilution. 

4. The kits used in this paper all detect WT SARS-CoV-2, but the current variants (omicron and its subvariants) have accumulated significant mutations that lead to immune-escape. Is it possible to determine which variant the patients were infected with? Or at least you should mention in discussion about the variants. And for the humoral response, it is always better to measure neutralization instead of ELISA (binding). 

Minor:

1. S1 is not equivalent to RBD. (S1 has NTD in addition to RBD) Please substitute S1 with RBD. 

2. please pay attention to the formating of some paragraphs. Correct any spelling/ grammatical mistakes (as attached)

Author Response

  1. Please provide a data plot (with individual values of scattered plot and mean/std) for Table.1 on the antibody levels, CD4 and CD8 T cells (can be in supplementary information). The data in this table looks questionable, especially the IgM level, where average (64) exceeds maximum (52)

Thank you for this comment. We have added a figure with the violin plot as suggested (supplementary figure 1 and 2, respectively). With respect to IgM levels, median value of the 20 CVID patients was 17. There were 3 patients with high serum IgM, which were suspected of hyper-IgM but did not show IgM genetic variants. We have added this information to the Results section (Page 4, Lines 1-2). 

  1. Use a high-resolution (better .svg file, or directly exported as 600dpi jpeg/png format) image for all your figures. The y-axis of your figure should be IFN-γ not γIFN

Thank you. We have replaced all figures for better quality ones. We have also substituted γIFN for IFN-γ. 

  1. The IFN-γ level of your samples look questionable. In this paper, 10.1016/j.jcv.2022.105098   they used the same kit and the result looks more plausible (max. 10,000 or more). The maximum value of your IFN-γ looks similar, with a plateau of ~2000, which might be due to inappropriate dilution. 

Thank you so much for highlighting this relevant point. The values of IFN-γ were obtained from subtracting from each patient the background IFN-γ stimulation (blank values). We did not dilute the supernatants with IFN-γ concentrations above 2,000 mIU/ml. We have acknowledged this limitation in the discussion, while it does not make any difference in the results shown. 

  1. The kits used in this paper all detect WT SARS-CoV-2, but the current variants (omicron and its subvariants) have accumulated significant mutations that lead to immune-escape. Is it possible to determine which variant the patients were infected with? Or at least you should mention in discussion about the variants. And for the humoral response, it is always better to measure neutralization instead of ELISA (binding).

Thank you for this comment. We did not sequence the specific variants in our patients. However, the Microbiology Dept. of our hospital performed sequencing of variants in random COVID patients to ascertain the predominant SARS-CoV-2 variant at every wave. At the time of our study, omicron variant was the predominant variant. We have added this information and extend on this, as suggested by this reviewer. We agree that specific neutralization studies are much more informative. However, we have no availability to assays measuring neutralizing antibodies against SARS-CoV-2. Further studies are needed to understand whether high IFN-γ and antibody titers were correlated with a good protection against SARS-CoV-2 infection.  

Minor: 

  1. S1 is not equivalent to RBD. (S1 has NTD in addition to RBD) Please substitute S1 with RBD. 2. please pay attention to the formating of some paragraphs. Correct any spelling/ grammatical mistakes (as attached).

We appreciate these constructive comments and have made the suggested changes. 

We appreciate your careful reading and your insightful interpretation of our results and suggestions. We have carefully reviewed and build on each of the comments, marked in yellow in the manuscript. 

Reviewer 2 Report

Mohamed and co-authors performed SARS-CoV-2 immunological study in patients with primary immunodeficiency, secondary immunodeficiency and healthy controls. The study targets specific immunocompromised individuals. 

[1] The main groups of interests (i.e., PID and SID) are limited in study group size. This likely result in lack of power in detection of differences between PID and SID and healthy controls. The authors clearly acknowledge this limitation in their discussion. Consequently, to make the findings of this manuscript stronger, follow-up data of this cohort is needed. 

[2] An important limitation of this study in my opinion is that only the antibody and cellular immune response against SARS-CoV-2 was investigated at 3-6 weeks post 2nd dose for the three groups. With most individuals already receiving 3 to 5 vaccinations in total, the present information has become less relevant and up-to-date. The authors state at the end of the manuscript that “further analysis is ongoing”, e.g. third vaccine dose. I believe these results should already be available by this time. I thus strongly recommend that the authors include the latest information of this ongoing analysis in the current manuscript as well.

At this moment, the presented data derived from two years ago is outdated. 

Minor comments:

[3] Abstract: Please specify more clearly in the text whether “Adequate specific cellular responses were observed in 18 out of 20 ….. 74 out of 81 (96%) HC.” is after vaccination. 

[4] Introduction – 2nd row: The wording “specific groups of risk” is unclear. Please specify whether authors refer to patient groups and what risk is mentioned (e.g., severe disease). 

[5] 3.2 SARS-CoV-2 History – 3rd row: “vaccine schedule” 

[6] 3.5: I suggest presenting no decimals in e.g. anti-spike IgG levels as these details are not relevant and reliable. 

Author Response

Reviewer 2 

[1] The main groups of interests (i.e., PID and SID) are limited in study group size. This likely result in lack of power in detection of differences between PID and SID and healthy controls. The authors clearly acknowledge this limitation in their discussion. Consequently, to make the findings of this manuscript stronger, follow-up data of this cohort is needed.  

[2] An important limitation of this study in my opinion is that only the antibody and cellular immune response against SARS-CoV-2 was investigated at 3-6 weeks post 2nd dose for the three groups. With most individuals already receiving 3 to 5 vaccinations in total, the present information has become less relevant and up-to-date. The authors state at the end of the manuscript that “further analysis is ongoing”, e.g. third vaccine dose. I believe these results should already be available by this time. I thus strongly recommend that the authors include the latest information of this ongoing analysis in the current manuscript as well. At this moment, the presented data derived from two years ago is outdated.  

Thank you so much for highlighting these points.  

We have acknowledged the limitations of our study in terms of sample size. However, the comparison between PID and SID remains and interesting value of our study. 

We have updated for all PID and SID patients follow-up of COVID-19 infection and booster doses. 

We are currently evaluating specific cellular and humoral responses in immunocompromised patients after booster doses in other risk groups. We have recently published data of adaptive immune response in cancer patients (10.3389/fonc.2022.975980). 

Minor comments: 

[3] Abstract: Please specify more clearly in the text whether “Adequate specific cellular responses were observed in 18 out of 20 ….. 74 out of 81 (96%) HC.” is after vaccination.  

We appreciate this constructive comment and have made the changes suggested. 

[4] Introduction – 2nd row: The wording “specific groups of risk” is unclear. Please specify whether authors refer to patient groups and what risk is mentioned (e.g., severe disease).  

Thank you for this comment. Specific group of risk patients refers to people with underlying health conditions, such as hypertension, diabetes, cardiovascular disease, chronic respiratory disease and weakened immune systems. 

[5] 3.2 SARS-CoV-2 History – 3rd row: “vaccine schedule”  

Thank you for this comment. It refers to the SARS-CoV-2 vaccine schedule in Spain. 

[6] 3.5: I suggest presenting no decimals in e.g. anti-spike IgG levels as these details are not relevant and reliable.  

We absolutely agree and have removed decimals accordingly. 

Reviewer 3 Report

The paper Specific Cellular and Humoral Immune Responses to the Neoantigen S1 of SARS-CoV-2 in Patients with Primary and Secondary Immunodeficiency and Healthy Donorsprovided the insignt into the immune response of SARS-CoV 2 vaccine or infection in Immunodeficiency and Healthy Donors

However, as we known the immune response strenghth was different in individuals with SARS-CoV 2 infections or vaccination, and there no clearly analysis the difference of the two kind population.

Second, what is ARS-CoV-2 IGRA stimulation tube set used for the Quan-T-Cell SARS-CoV-2 kit, as we know the stimulation and activation of CD 4 or CD 8 cell was the peptide pool, and the T cells is not the only cells that secret IFN-r.

Author Response

Reviewer 3 

However, as we known the immune response strength was different in individuals with SARS-CoV 2 infections or vaccination, and there no clearly analysis the difference of the two kind population. 

Thank you for this comment. We studied 10 PID patients before vaccination to evaluate basal IFN-γ production and we saw that 4 patients had COVID-19 prior to vaccination. Moreover, we evaluated adaptive immune response in hybrid healthy controls seeing that this subgroup presented the highest IFN-γ titers. 

Second, what is SARS-CoV-2 IGRA stimulation tube set used for the Quan-T-Cell SARS-CoV-2 kit, as we know the stimulation and activation of CD 4 or CD 8 cell was the peptide pool, and the T cells is not the only cells that secret IFN-r. 

Thank you so much for highlighting this point. IFN-γ production is largely restricted to T lymphocytes and natural killer (NK) cells. It was described that COVID-19 patients produce SARS-CoV-2-reactive IFN-γ-releasing T cells. Petruccioli E, Najafi Fard S, Navarra A, Petrone L, Vanini V, Cuzzi G, Gualano G, Pierelli L, Bertoletti A, Nicastri E, Palmieri F, Ippolito G, Goletti D. Exploratory analysis to identify the best antigen and the best immune biomarkers to study SARS-CoV-2 infection. J Transl Med. 2021 Jun 26;19(1):272. doi: 10.1186/s12967-021-02938-8. PMID: 34174875; PMCID: PMC8235902. 

We appreciate your careful reading and your insightful interpretation of our results and suggestions. We have carefully reviewed and build on each of the comments, marked in yellow in the manuscript. 

Round 2

Reviewer 2 Report

Unfortunately, the authors did not address my questions and concerns sufficiently. Especially regarding the comment on presenting the follow-up data an unclear response was given that the authors investigated the effect after booster vaccination in other risk groups, which contradicts with statements in the conclusion. In that case, altogether, as said below the data of >2 years ago (in an era prior to booster vaccinations) have become outdated in my opinion. Additional experiments are needed. 
